# Percutaneous Radiofrequency Ablation with or without Chemolipiodolization for Hepatocellular Carcinoma: A Propensity-Score-Matched Analysis

**DOI:** 10.3390/jcm11061483

**Published:** 2022-03-08

**Authors:** Kota Takaki, Masahito Nakano, Kazuta Fukumori, Yoichi Yano, Yuki Zaizen, Takashi Niizeki, Kotaro Kuwaki, Masaru Fukahori, Takahiko Sakaue, Sohei Yoshimura, Mika Nakazaki, Takuji Torimura

**Affiliations:** 1Division of Gastroenterology, Department of Medicine, Kurume University School of Medicine, Kurume 830-0011, Japan; takaki_kouta@med.kurume-u.ac.jp (K.T.); fukumori_kazuta@kurume-u.ac.jp (K.F.); zaizen_yuki@med.kurume-u.ac.jp (Y.Z.); niizeki_takashi@kurume-u.ac.jp (T.N.); kuwaki_koutarou@kurume-u.ac.jp (K.K.); fukahori_masaru@med.kurume-u.ac.jp (M.F.); sakaue_takahiko@med.kurume-u.ac.jp (T.S.); yoshimura_souhei@med.kurume-u.ac.jp (S.Y.); nomiyama_mika@med.kurume-u.ac.jp (M.N.); tori@med.kurume-u.ac.jp (T.T.); 2Division of Gastroenterology, Department of Medicine, Japan Community Health Care Organization, Saga Central Hospital, Saga 849-8522, Japan; syano@kumin.ne.jp; 3Division of Gastroenterology, Department of Medicine, Omuta City Hospital, Omuta 836-8567, Japan

**Keywords:** liver cancer, risk factors, chemoembolization, radical treatment, transarterial, survival

## Abstract

Chemolipiodolization (CL) is less invasive than transarterial chemoembolization (TACE) for managing hepatocellular carcinoma (HCC) because it helps avoid embolization. However, the treatment outcomes of percutaneous radiofrequency ablation (PRFA) with or without CL for HCC remain unclear. Herein, we compared the prognostic factors for overall survival (OS) following PRFA with or without CL for HCC using propensity-score-matched analysis. A total of 221 patients with HCC treated with PRFA at Saga Central Hospital between April 2004 and October 2020, with or without CL, were enrolled. No significant difference was observed in OS between PRFA with and without CL cohorts (median survival time (MST): 4.5 vs. 5.4 years; *p* = 0.0806). To reduce the confounding effects of 12 variables, we performed propensity-score-matched analysis to match patients treated with PRFA with or without CL. No significant difference was observed in OS between PRFA with and without CL cohorts (MST: 4.0 vs. 3.6 years; *p* = 0.5474). After stratification according to tumor size, no significant difference was observed in OS for patients with tumor size ≥20 mm between PRFA with and without CL cohorts (MST: 3.5 vs. 3.4 years; *p* = 0.8236). PRFA with CL was not a significant prognostic factor in both univariate and multivariate analyses (*p* = 0.5477 and 0.9600, respectively). Our findings suggest that PRFA with CL does not demonstrate more favorable prognosis than PRFA without CL for HCC, regardless of tumor size.

## 1. Introduction

In 2018, liver cancer was the sixth most prevalent cancer and the fourth leading cause of cancer-related deaths worldwide, with an estimated 841,000 new cases and 782,000 deaths [1,2,3,4]. Liver cancer includes hepatocellular carcinoma (HCC), accounting for 75–85% of all liver cancer cases [1,2]. HCC may be cured by radical means through hepatic resection, radiofrequency ablation (RFA), or liver transplantation [5,6].

Percutaneous RFA (PRFA) is a simple and minimally invasive treatment option for HCC [7]. Moreover, combination therapy of PRFA and transarterial chemoembolization (TACE) can improve survival in patients with HCC [7]. In a prospective randomized trial, RFA combined with TACE was superior to RFA alone in improving the survival of patients with HCC [8]. Theoretically, it is possible to diminish the cooling effect of blood and consequently expand the area of ablation by performing TACE before PRFA. Recently, there has been accumulating evidence regarding the efficacy of PRFA with TACE for HCC treatment [9,10,11].

Chemolipiodolization (CL) is a standard procedure performed before embolization in TACE, which is common for HCC treatment [12,13,14,15]. However, several studies have reported that CL plays a major role and embolization does not improve survival [16,17], whereas other studies have drawn contrary conclusions [18]. CL is also performed before hepatic resection, and preoperative CL of the entire liver has been reported to be effective in reducing the incidence of postoperative recurrence and prolonging survival in patients with resectable HCC [19]. Thus, the evaluation of CL for HCC remains controversial.

Nevertheless, CL for HCC is considered to have an advantage over TACE because it has fewer effects on the body, since no embolization is performed. However, the treatment outcomes of PRFA with or without CL for HCC remain unclear. Therefore, in this study, we aimed to determine the prognostic effects of PRFA with or without CL and the associated overall survival (OS) duration in HCC. We performed propensity-score-matched analysis to reduce the effects of confounders in view of this objective.

## 2. Materials and Methods

### 2.1. Diagnosis

HCC was confirmed histologically or diagnosed using non-invasive criteria according to the European Association for the Study of the Liver [20]. Intrahepatic lesions and vascular invasion were diagnosed using a combination of imaging techniques, such as contrast-enhanced computed tomography, magnetic resonance imaging, ultrasonography, and digital subtraction angiography. Additionally, serum alpha-fetoprotein (AFP) and des-gamma-carboxy prothrombin (DCP) levels were measured for up to one month before treatment initiation. The presence of intra-abdominal metastases was determined using abdominal computed tomography, magnetic resonance imaging, and ultrasonography, which were performed to evaluate intrahepatic lesions. Liver function was assessed using the Child–Pugh classification and albumin-bilirubin (ALBI) score [21]. The tumor stage was determined according to the Barcelona Clinic Liver Cancer (BCLC) staging classification [22,23].

### 2.2. Patients Receiving PRFA

Patients with HCC underwent PRFA at Saga Central Hospital between April 2004 and October 2020. Among them, we excluded those undergoing their second or later PRFA; therefore, we only enrolled consecutive patients who received PRFA as the first treatment. Ultrasound-guided PRFA was performed one week after CL. Specifically, we used the Cool-Tip RF system (Covidien, Boulder, CO, USA) that utilizes internally cooled electrodes for ablation through an internal cooling device, thus reducing the impedance. We also used needle electrodes of various lengths depending on the tumor size. Additionally, we heated the HCC tissue to 70−80 °C until the impedance markedly increased, which created frictional heat, causing the death of tumor cells. Since tumor ablation depends on the impedance of the tissue and is proportionate to the distance from electrodes, an appropriate needle was selected.

### 2.3. Patients Receiving CL

Among the enrolled patients who received PRFA, some patients received CL without embolization on the day before PRFA. After conventional visceral angiography, CL was performed according to the procedure described in previous studies [15,16,19], by introducing an angiographic catheter into the feeding artery of HCC using Seldinger’s technique. An angiographic survey of abdominal vessels such as the superior mesenteric artery and common hepatic vessels was performed to assess the arterial blood supply to the liver. Epirubicin was mixed with a water-soluble contrast medium and sterile water for injection and was thoroughly mixed with lipiodol. The injection was stopped either at the point of near stasis within the feeding artery of HCC or after the entire amount of the agent was administered. We assessed the range of injection rate for the tumor using lipiodol after CL. Some studies have revealed that adjuvant CL can reduce the risk of intrahepatic metastasis recurrence, albeit not multicentric carcinogenesis [24].

### 2.4. Treatment Outcome

The treatment outcome was OS, defined as the time from PRFA initiation with or without CL to the date of death or the patient’s last follow-up, whichever occurred first.

### 2.5. Statistical Analyses

The following baseline patient characteristics were analyzed using descriptive statistical methods: age, tumor size, albumin level, total bilirubin level, ALBI score, prothrombin time, AFP, and DCP. Comparisons of these continuous variables were performed using the *t*-test, and comparisons of categorical variables such as sex, etiology, Child–Pugh class, and BCLC stage were calculated using the chi-square test. Results are expressed as mean ± standard deviation (SD) and median (range) or *n* (%). Survival curves were constructed using the Kaplan–Meier analysis with the log-rank test. A *p*-value of <0.05 was considered statistically significant. JMP software (SAS Institute, Inc., Cary, NC, USA), version 15, was used for all statistical analyses.

## 3. Results

### 3.1. Patient Characteristics

Overall, 501 patients with HCC were treated with PRFA, of whom 280 undergoing second or later PRFA were excluded. Finally, 221 patients who received PRFA as the first treatment were enrolled.

Table 1 presents the characteristics of patients who were diagnosed with HCC and received PRFA either with (*n* = 76) or without (*n* = 145) CL. A higher proportion of patients had a large tumor size (*p* < 0.0001) in the PRFA with CL cohort, whereas a higher proportion of patients had BCLC stage 0 (*p* < 0.0001) in the PRFA without CL cohort. There were no significant differences in age, sex, etiology, Child–Pugh class, albumin level, total bilirubin level, ALBI score, prothrombin time, AFP level, and DCP level between the PRFA with and without CL cohorts.

### 3.2. Survival Outcomes

Figure 1 depicts the results of the Kaplan–Meier analysis; OS was assessed with the log-rank test in the PRFA with and without CL cohorts. The median survival time (MST) was 4.5 years in the PRFA with CL cohort (red line; *n* = 76) and 5.4 years in the PRFA without CL cohort (blue line; *n* = 145) (*p* = 0.0806). OS did not differ significantly between the PRFA with and without CL cohorts.

### 3.3. Propensity-Score-Matched Analysis

To reduce confounding effects, we performed propensity-score-matched analysis to match patients treated with PRFA with CL (*n* = 76) and those treated without CL (*n* = 145) [25,26]. The following 12 variables related to the prognosis of HCC were considered at the start of the follow-up: age, sex, etiology, Child–Pugh class, tumor size, BCLC stage, albumin level, total bilirubin level, ALBI score, prothrombin time, AFP level, and DCP level. The propensity scores (mean ± SD) of the patients treated with PRFA with or without CL were 0.4036 ± 2.3596 and −2.2509 ± 4.5160, respectively. We used these propensity scores to conduct one-to-one nearest neighbor matching within a caliper of 0.20, as previous studies have revealed this SD percentage of the logit of the propensity score to be generally suitable for propensity-score-matched analysis [27]. Based on the propensity-score-matched analysis results, 108 patients were selected (PRFA with CL, *n* = 54; PRFA without CL, *n* = 54). Moreover, the propensity scores (mean ± SD) of patients treated with PRFA with or without CL were −0.3854 ± 0.9998 and −0.4535 ± 0.9328, respectively.

### 3.4. Patient Characteristics after Propensity-Score-Matched Analysis

Table 2 presents the characteristics of 108 patients who were diagnosed with HCC and underwent PRFA with CL (*n* = 54) or without CL (*n* = 54), assessed using propensity-score-matched analysis. No significant differences were observed in any variables between the PRFA with and without CL cohorts on propensity-score-matched analysis.

### 3.5. Survival Outcomes after Propensity-Score-Matched Analysis

Figure 2 depicts the results of the Kaplan–Meier analysis; OS was assessed with the log-rank test between the PRFA with and without CL cohorts after propensity-score-matched analysis. MST was 4.0 years in the PRFA with CL cohort (red line; *n* = 54) and 3.6 years in the PRFA without CL cohort (blue line; *n* = 54) (*p* = 0.5474). After propensity-score-matched analysis, OS was not significantly different between the PRFA with and without CL cohorts.

### 3.6. Survival Outcomes after Propensity-Score-Matched Analysis Post Stratification of Patients According to the Tumor Size

Figure 3 depicts the results of the Kaplan–Meier analysis of OS after stratification of patients according to the tumor size. This was assessed with the log-rank test to compare the PRFA with and without CL cohorts in patients with tumors sized ≥20 mm after propensity-score-matched analysis. MST was 3.5 years in the PRFA with CL cohort (red line; *n* = 26) and 3.4 years in the PRFA without CL cohort (blue line; *n* = 24) (*p* = 0.8236). After propensity-score-matched analysis, OS was not significantly different between the cohorts for patients with tumors sized ≥20 mm.

### 3.7. Univariate and Multivariate Analyses of OS after Propensity-Score-Matched Analysis

Table 3 presents the results of univariate and multivariate analyses of OS obtained using propensity-score-matched analysis (*n* = 108). Univariate analysis of OS revealed four variables as significant prognostic factors: Child–Pugh class (*p* = 0.0012), ALBI score (*p* < 0.0001), AFP level (*p* = 0.0032), and DCP level (*p* = 0.0119). Multivariate analyses of OS identified two variables as independent significant prognostic factors: Child–Pugh class (*p* = 0.0245) and DCP level (*p* = 0.0442). PRFA with CL was not a significant prognostic factor in both univariate and multivariate analyses following propensity-score matching.

## 4. Discussion

In this study, we observed that OS did not differ significantly between the PRFA with and without CL cohorts (Figure 1). However, in the PRFA without CL cohort, tumor factors, especially tumor size, were significantly better (Table 1). To reduce confounding effects, we performed propensity-score-matched analysis to match patients treated with PRFA with and without CL (Table 2). OS did not differ significantly between the two cohorts after propensity-score-matched analysis (Figure 2). Similarly, OS did not differ significantly between the PRFA with and without CL cohorts in patients with a tumor size of ≥20 mm after propensity-score-matched analysis (Figure 3). Our results suggest that PRFA with CL does not demonstrate more prolonged prognostic effects than PRFA without CL in HCC, regardless of the tumor size. Additionally, in both univariate and multivariate analyses, PRFA with CL was not a significant prognostic factor after propensity-score-matched analysis (Table 3).

There are several treatment strategies for HCC, such as PRFA and surgical resection. PRFA is an effective therapy for HCC when the tumor size is <3 cm in diameter and the number of tumors is <3 [7]. On the contrary, CL is an effective treatment option for unresectable HCC [28]. Similarly, PRFA with TACE is effective in HCC treatment [29]. PRFA combined with TACE is superior to PRFA alone in improving the survival of patients with HCC of <7 cm in size [8]. Percutaneous microwave coagulo-necrotic therapy with TACE can also be used to effectively treat HCC measuring >2 cm but <3 cm [30]. A meta-analysis also indicated that PRFA combined with TACE demonstrates higher tumor response rates and improved survival rates [31]. In PRFA, it is suggested that the heat diffusion within the tumor is affected by intratumoral septa and fibrosis [15]. Intratumoral septa are usually disrupted after TACE, facilitating heat distribution within the tumor. Conversely, in a propensity-score-matched study, embolization in TACE combined with PRFA could not improve the survival of patients with HCC according to the Milan criteria [15]. However, only a few studies have reported improved OS by PRFA with CL. Our study identified no significant value of adding CL to PRFA.

It is well known in general that PRFA (without CL) improves the prognosis for patients with HCC compared with TACE [7]. On the other hand, the comparison of PRFA with CL and TACE remains controversial. However, we reported that TACE combined with PRFA improved the prognosis of patients with HCC compared with TACE alone [32]. As a result, we think that PRFA with CL might improve the prognosis for patients with HCC compared with TACE. We have treated only a few patients with HCC with TACE during this study period at our hospital, because it is crucial that the feeding artery be selectively embolized during TACE; otherwise, there is a risk of liver failure due to the use of embolic material in the treatment of HCC that has spread widely. Therefore, to resolve this controversial point, a multicenter prospective study with a larger patient population should be conducted in the future.

The induction of the cooling effect may explain why there was no significant difference in the OS of patients who underwent PRFA with CL. However, a previous study revealed prolongation of survival in patients undergoing PRFA with TACE [8]. It is well known that blood vessels around the PRFA probe cause a cooling effect and decrease the coagulation size. Estimating the cooling effect before PRFA with a CT scan is difficult; thus, some previous studies have evaluated the cooling effect using a mathematical model [24]. Additional procedures, such as a Pringle maneuver or CL, are needed to avoid the cooling effect. Obstruction of the hepatic artery effectively reduces the cooling effect since the hepatic artery is the main source of blood supply to HCC; therefore, PRFA is sufficient to expand the area of necrosis after hepatic artery obstruction [33]. Since CL is a procedure that does not use embolic substances, it may not effectively reduce the cooling effect of the hepatic blood flow on PRFA, which diminishes the cauterization effect. The removal of embolization from the combination therapy of TACE and RFA is not important for improving the survival rate of patients with HCC [15]. There was no significant difference between PRFA with CL and PRFA without CL in our study. Therefore, this suggests the importance of embolization with PRFA.

In summary, PRFA in combination with CL could not improve OS in patients with HCC. The reason is that CL does not involve embolic substances, which does not reduce the cooling effect. In addition, CL increases medical expenses and the risk of complications such as bleeding and infections, including biloma and skin abscess.

This study has some limitations. First, this was a single center, retrospective study with a relatively small sample size (*n* = 221) of patients with HCC. Second, the treatment (PRFA with or without CL) was selected at the discretion of the chief physician, and patients were not randomized to the treatment. This resulted in a selection bias for patients with HCC. Third, therapeutic effects and adverse events in all cases could not be evaluated. Fourth, no further investigations were conducted after secondary treatment. Therefore, to overcome these limitations, a multicenter prospective study with a larger patient population should be conducted in the future.

## 5. Conclusions

This study demonstrated that PRFA with CL does not yield more prolonged prognostic effects than PRFA without CL on HCC. Our results suggest that PRFA with CL should not be employed for patients with HCC, regardless of the tumor size.

## Figures and Tables

**Figure 1 jcm-11-01483-f001:**
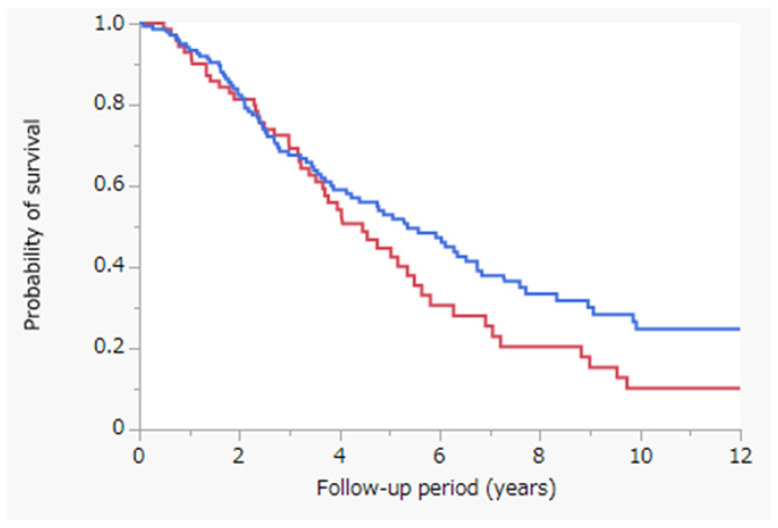
Kaplan–Meier analysis of OS with the log-rank test between the PRFA with and without CL cohorts. Red line, PRFA with CL cohort (*n* = 76), MST = 4.5 years; blue line, PRFA without CL cohort (*n* = 145), MST = 5.4 years; *p* = 0.0806. Abbreviations: OS, overall survival; PRFA, percutaneous radiofrequency ablation; CL, chemolipiodolization; MST, median survival time.

**Figure 2 jcm-11-01483-f002:**
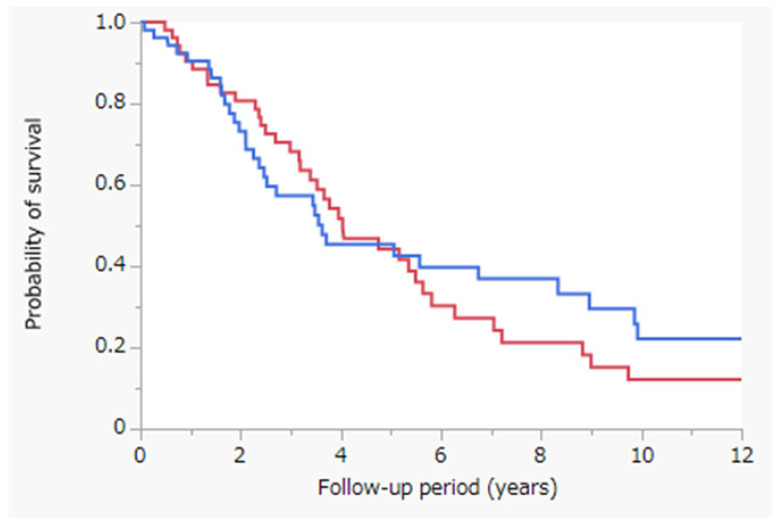
Kaplan–Meier analysis of OS with the log-rank test between the PRFA with and without CL cohorts following propensity-score-matched analysis. Red line, PRFA with CL cohort (*n* = 54), MST = 4.0 years; blue line, PRFA without CL cohort (*n* = 54), MST = 3.6 years; *p* = 0.5474. Abbreviations: OS, overall survival; PRFA, percutaneous radiofrequency ablation; CL, chemolipiodolization; MST, median survival time.

**Figure 3 jcm-11-01483-f003:**
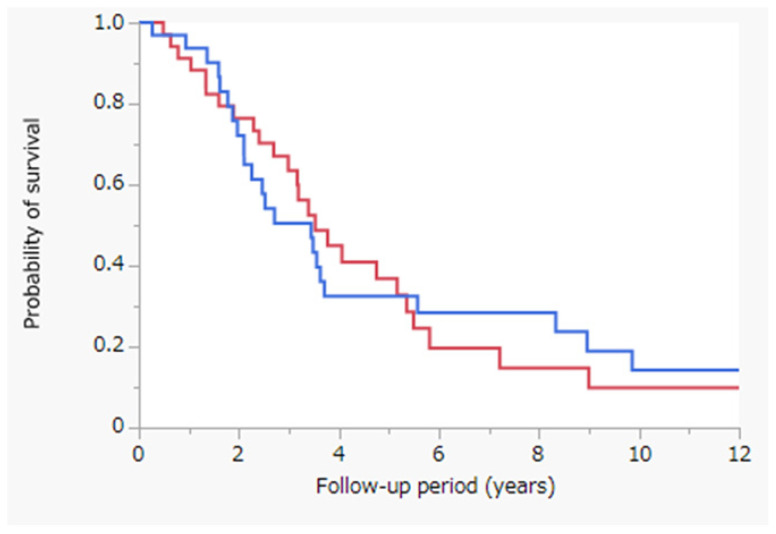
Kaplan–Meier analysis of OS with the log-rank test between the PRFA with and without CL cohorts in patients with a tumor size of ≥20 mm following propensity-score-matched analysis. Red line, PRFA with CL cohort in patients with a tumor size of ≥20 mm (*n* = 26), MST = 3.5 years; blue line, PRFA without CL cohort in patients with a tumor size of ≥20 mm (*n* = 24), MST = 3.4 years; *p* = 0.8236. Abbreviations: OS, overall survival; PRFA, percutaneous radiofrequency ablation; CL, chemolipiodolization; MST, median survival time.

**Table 1 jcm-11-01483-t001:** Patient characteristics (*n* = 221).

Variable	CL (+) (*n* = 76)	CL (−) (*n* = 145)	*p*-Value
Age (years)	73.1 ± 10.7 75.4 (32.8–88.6)	73.7 ± 8.3 75.2 (46.0–88.6)	0.6711
Sex (Male/Female)	52 (68%)/24 (32%)	80 (55%)/65 (45%)	0.0564
Etiology (HBV/HCV/Both negative)	4 (5%)/66 (87%)/6 (8%)	8 (5%)/129 (90%)/8 (5%)	0.7881
Child–Pugh class (A/B)	62 (82%)/14 (18%)	120 (83%)/25 (17%)	0.4766
Tumor size (mm)	25.3 ± 9.0 25 (9–48)	18.1 ± 5.6 17 (8–35)	<0.0001
BCLC stage (0/A/B)	20 (26%)/30 (40%)/26 (34%)	85 (59%)/50 (34%)/10 (7%)	<0.0001
Albumin (g/dL)	3.6 ± 0.5 3.7 (2.7–4.7)	3.7 ± 0.5 3.7 (2.5–4.8)	0.4492
Total bilirubin (mg/dL)	1.0 ± 0.4 0.8 (0.3–2.5)	0.9 ± 0.5 0.8 (0.2–3.2)	0.5275
ALBI score	−2.30 ± 0.45 −2.32 (−3.38–1.29)	−2.36 ± 0.47 −2.40 (−3.52–1.21)	0.3223
Prothrombin time (%)	77.1 ± 12.2 75.2 (44.5–108.7)	78.4 ± 12.7 78.7 (46.9–108.3)	0.4631
AFP (ng/mL)	296 ± 1009 32 (2–8400)	118 ± 39 15 (1–5019)	0.0758
DCP (mAU/mL)	905 ± 5042 47 (2–42,500)	144 ± 443 25 (8–3930)	0.0751

CL, chemolipiodolization; HBV, hepatitis B virus; HCV, hepatitis C virus; BCLC, Barcelona Clinic Liver Cancer; ALBI, albumin-bilirubin; AFP, alpha-fetoprotein; DCP, des-gamma-carboxy prothrombin.

**Table 2 jcm-11-01483-t002:** Patient characteristics following propensity-score-matched analysis (*n* = 108).

Variable	CL (+) (*n* = 54)	CL (−) (*n* = 54)	*p*-Value
Age (years)	73.2 ± 11.4 75.7 (32.9–88.6)	72.8 ± 7.8 74.5 (55.2–88.6)	0.8496
Sex (Male/Female)	36 (67%)/18 (33%)	37 (69%)/17 (31%)	0.8371
Etiology (HBV/HCV/Both negative)	3 (6%)/48 (88%)/3 (6%)	3 (6%)/46 (87%)/5 (7%)	0.7624
Child–Pugh class (A/B)	45 (83%)/9 (17%)	42 (78%)/12 (22%)	0.4658
Tumor size (mm)	22.6 ± 7.6 22 (9–37)	21.9 ± 6.2 21 (11–35)	0.5990
BCLC stage (0/A/B)	18 (33%)/24 (45%)/12 (22%)	17 (31%)/27 (50%)/10 (19%)	0.8241
Albumin (g/dL)	3.6 ± 0.5 3.8 (2.5–4.5)	3.59 ± 0.5 3.6 (2.5–4.8)	0.7066
Total bilirubin (mg/dL)	0.9 ± 0.4 0.8 (0.3–2.1)	0.9 ± 0.4 0.9 (0.4–2.2)	0.4172
ALBI score	−2.32 ± 0.43 −2.33 (−3.20–1.37)	−2.27 ± 0.47 −2.24 (−3.52–1.21)	0.5726
Prothrombin time (%)	77.9 ± 12.8 78.3 (44.5–108.7)	76.6 ± 14.4 74.8 (46.9–106.1)	0.6370
AFP (ng/mL)	154 ± 279 32 (2–1170)	124 ± 312 22 (4–1628)	0.5996
DCP (mAU/mL)	286 ± 700 46 (2–3310)	282 ± 675 34 (8–3930)	0.9781

Notes: Results are expressed as mean ± standard deviation and median (range) or *n* (%). CL, chemolipiodolization; HBV, hepatitis B virus; HCV, hepatitis C virus; BCLC, Barcelona Clinic Liver Cancer; ALBI, albumin-bilirubin; AFP, alpha-fetoprotein; DCP, des-gamma-carboxy prothrombin.

**Table 3 jcm-11-01483-t003:** Results of univariate and multivariate analyses of OS following propensity-score-matched analysis (*n* = 108).

Variable	Univariate Analysis		Multivariate Analysis	
	HR (95% CI)	*p*-Value	HR (95% CI)	*p*-Value
Age (≥75.2 years)	1.237 (0.771–1.993)	0.3767	1.043 (0.614–1.785)	0.8752
Sex (Male)	1.249 (0.758–2.057)	0.3757	1.581 (0.906–2.759)	0.0995
Etiology (HCV)	1.946 (0.980–4.447)	0.0574	1.512 (0.697–3.655)	0.3063
Child–Pugh class (B)	3.126 (1.616–5.720)	0.0012	2.283 (1.117–4.507)	0.0245
Tumor size (≥22 mm)	1.541 (0.963–2.494)	0.0712	1.338 (0.690–2.804)	0.4001
BCLC stage (A or B)	1.477 (0.888–2.566)	0.1356	0.928 (0.411–1.994)	0.8525
ALBI score (≥−2.31)	3.097 (1.750–5.488)	<0.0001	1.370 (0.749–2.496)	0.3036
AFP (≥27.6 ng/mL)	2.059 (1.274–3.361)	0.0032	1.669 (0.971–2.907)	0.0638
DCP (≥43 mAU/mL)	1.862 (1.146–3.062)	0.0119	1.705 (1.013–2.903)	0.0442
Treatment (PRFA with CL)	1.155 (0.721–1.851)	0.5477	1.012 (0.615–1.667)	0.9600

Notes: All cutoff values were set as the median. OS, overall survival; HR, hazard ratio; CI, confidence interval; HCV, hepatitis C virus; BCLC, Barcelona Clinic Liver Cancer; ALBI, albumin-bilirubin; AFP, alpha-fetoprotein; DCP, des-gamma-carboxy prothrombin; CL, chemolipiodolization; PRFA, percutaneous radiofrequency ablation.

## Data Availability

The data that support the findings of this study are available from the corresponding author, Nakano M, on reasonable request.

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
