# Peer review of "Percutaneous Radiofrequency Ablation with or without Chemolipiodolization for Hepatocellular Carcinoma: A Propensity-Score-Matched Analysis"

_jcm, 2022, doi:10.3390/jcm11061483_

Round 1

Reviewer 1 Report

The authors presented an important study comparing PRFA with and without CL. The study might provide additional benefit of PRFA in the management of HCC. However, there are some points that need to be addressed:

-  Although propensity score-matched analysis was performed, the two groups of PRFA with and without CL might receive different additional treatment including targeted therapy that might influence the overall survivals. Please address this concern.

- Preoperative clinical variables including BMI, metabolic comorbidities, performance score might also influence the outcome that have not been fully explained in this study.

- In the discussion, please add the limitations of the study including the study design that was not an RTC

- Please also discuss the comparison of PRFA with and without CL with TACE.

Author Response

The authors presented an important study comparing PRFA with and without CL. The study might provide additional benefit of PRFA in the management of HCC. However, there are some points that need to be addressed:

-  Although propensity score-matched analysis was performed, the two groups of PRFA with and without CL might receive different additional treatment including targeted therapy that might influence the overall survivals. Please address this concern.

Response: Thank you for your important comment. Unfortunately, the additional treatment provided to all patients could not be evaluated since electronic medical records were introduced in our hospital only on February 1, 2017. Therefore, we could evaluate the additional treatment of only 15 patients, retrospectively. The breakdown of the additional treatment was as follows: six patients received re-CL, five received re-PRFA, three received immunotherapy with atezolizumab and bevacizumab, and two received molecular targeted therapy with sorafenib (overlapped). Five patients did not receive any further therapy.

- Preoperative clinical variables including BMI, metabolic comorbidities, performance score might also influence the outcome that have not been fully explained in this study.

Response: Thank you for your important comment. For the same reason as above, we were able to evaluate preoperative clinical variables including BMI, metabolic comorbidities, and performance status (PS) of only 15 patients retrospectively. BMI was 25.0 ± 5.6 kg/m2 (mean ± SD). Four patients had metabolic comorbidities (diabetes mellitus), while 11 patients did not. PS was 0 in all patients.

- In the discussion, please add the limitations of the study including the study design that was not an RTC

Response: Thank you for your suggestion. We mentioned this point as a study limitation in the “Discussion” section as follows: “The treatment (PRFA with or without CL) was selected at the discretion of the chief physician, and patients were not randomized to the treatment.”

- Please also discuss the comparison of PRFA with and without CL with TACE.

Response: Thank you for your suggestion. We mentioned this point in the “Discussion” section as follows: “It is well known in general that PRFA (without CL) improves the prognosis for patients with HCC compared with TACE [7]. On the other hand, the comparison of PRFA with CL and TACE remains controversial. However, we reported that TACE combined with PRFA improved the prognosis of patients with HCC compared with TACE alone [32]. As a result, we think that PRFA with CL might improve the prognosis for patients with HCC compared with TACE. We have treated only a few patients with HCC with TACE during this study period at our hospital, because it is crucial that the feeding artery be selectively embolized during TACE; otherwise, there is a risk of liver failure due to the use of embolic material in the treatment of HCC that has spread widely. Therefore, to resolve this controversial point, a multicenter prospective study with a larger patient population should be conducted in the future.”

  1. Shimose, S.; Tanaka, M.; Iwamoto, H.; Niizeki, T.; Shirono, T.; Aino, H.; Noda, Y.; Kamachi, N.; Okamura, S.; Nakano, M.; et al. Prognostic impact of transcatheter arterial chemoembolization (TACE) combined with radiofrequency ablation in patients with unresectable hepatocellular carcinoma: Comparison with TACE alone using decision-tree analysis after propensity score matching. Hepatology research : the official journal of the Japan Society of Hepatology 2019, 49, 919-928, doi:10.1111/hepr.13348.

Reviewer 2 Report

Authors aimed to compare prognostic factors for overall survival (OS) following PRFA with or without     

CL for HCC using propensity score-matched analysis. There are several points to be addressed.

1) What are the criteria for CL ?

2) Is there any reason not using embolic material during CL?

3) Please, show the data regarding PFS and radiological response, i.e the rate of complete remission.

Author Response

Authors aimed to compare prognostic factors for overall survival (OS) following PRFA with or without     

CL for HCC using propensity score-matched analysis. There are several points to be addressed.

  • What are the criteria for CL?

Response: Thank you for your important comment. We decided TACE or CL as the treatment based on the range of the HCC and/or extent of blood supply to the HCC. Only when we could selectively identify the feeding artery did we perform TACE. On the other hand, when we had to inject the anti-cancer agent into the feeding artery of HCC which had spread widely, we performed CL in many cases.

  • Is there any reason not using embolic material during CL?

Response: Thank you for your important comment. CL involves injection of the anti-cancer agent into the feeding artery of HCC; therefore, it is advantageous in treating HCC that has spread widely. However, there is a risk of liver failure while using embolic material during CL for HCC that has spread widely; therefore, we did not use embolic material during CL.

  • Please, show the data regarding PFS and radiological response, i.e the rate of complete remission.

Response: Thank you for raising this pertinent point. Unfortunately, the therapeutic effect in all cases could not be evaluated since electronic medical records were introduced in our hospital only on February 1, 2017. Therefore, we were able to evaluate the radiological response of only 15 patients based on their imaging findings following PRFA, retrospectively. We have re-analyzed these cases and attached a figure on PFS in 15 patients for your kind reference. The figure presents the results of the Kaplan–Meier analysis of PFS using the log-rank test between the PRFA with and without CL cohorts among 15 patients. The median survival time was 1.4 years in the PRFA with CL cohort (red line; n = 4) and did not reach the PRFA without CL cohort (blue line; n = 11) (p = 0.0523). The rate of complete remission was 93%.

Round 2

Reviewer 2 Report

Authros addressed raised points well.